# Optimization of a Sustainable Protocol for the Extraction of Anthocyanins as Textile Dyes from Plant Materials

**DOI:** 10.3390/molecules26226775

**Published:** 2021-11-09

**Authors:** Elisa Gecchele, Stefano Negri, Anna Cauzzi, Anna Cuccurullo, Mauro Commisso, Alessia Patrucco, Anastasia Anceschi, Giorgio Zaffani, Linda Avesani

**Affiliations:** 1Department of Biotechnology, University of Verona, Strada Le Grazie 15, 37134 Verona, Italy; elisa.gecchele@univr.it (E.G.); stefano.negri@univr.it (S.N.); anna.cauzzi@hotmail.com (A.C.); anna.cuccurullo@univr.it (A.C.); mauro.commisso@univr.it (M.C.); 2CNR-STIIMA, Italian National Research Council, Institute of Intelligent Industrial Technologies and Systems for Advanced Manufacturing, Corso G. Pella 16, 13900 Biella, Italy; alessia.patrucco@stiima.cnr.it (A.P.); anastasia.anceschi@stiima.cnr.it (A.A.); 3Cooperativa Sociale Cercate, Via Bramante 15, 37134 Verona, Italy; g.zaffani@ecofarm.storti.com

**Keywords:** anthocyanin, red chicory, green economy, natural color dye, design-of-experiments, sustainable protocol

## Abstract

Anthocyanins are the largest group of polyphenolic pigments in the plant kingdom. These non-toxic, water-soluble compounds are responsible for the pink, red, purple, violet, and blue colors of fruits, vegetables, and flowers. Anthocyanins are widely used in the production of food, cosmetic and textile products, in the latter case to replace synthetic dyes with natural and sustainable alternatives. Here, we describe an environmentally benign method for the extraction of anthocyanins from red chicory and their characterization by HPLC-DAD and UPLC-MS. The protocol does not require hazardous solvents or chemicals and relies on a simple and scalable procedure that can be applied to red chicory waste streams for anthocyanin extraction. The extracted anthocyanins were characterized for stability over time and for their textile dyeing properties, achieving good values for washing fastness and, as expected, a pink-to-green color change that is reversible and can therefore be exploited in the fashion industry.

## 1. Introduction

Colored textile and leather materials have been produced using natural dyes since prehistoric times, but the development of inexpensive and readily available synthetic colorants has left natural dyes confined to small niche markets. However, current trends suggest a resurgence in the demand for natural dyes as consumers become more environmentally conscious and seek products with eco-friendly labels. Indeed, all synthetic dyes derived from petrochemical sources require the use of manufacturing processes involving harmful chemicals and solvents. Some toxic chemicals are carried over into the product, resulting in health problems including a rising incidence of skin allergies [1], and others are released into the environment [2]. In contrast, natural dyes can be extracted from renewable sources such as microbes and plants, and tend to be more biodegradable [3]. Plant-derived dyes occur in a wide palette of colors, and there is already a large body of knowledge based on thousands of years of practice in extraction and dyeing [4].

The use of agricultural waste for the extraction of colorants embraces the concept of the green economy, changing a waste stream into a valuable resource. However, the practical application of this concept requires optimized extraction protocols that maximize the final colorant yields and their stability over time, while ensuring that the extraction process remains environmentally benign. Such sustainable extraction processes should minimize energy consumption, allow the use of alternative solvents and renewable natural products, and ensure the production of safe, high-quality extracts [5]. Moreover, the standardization of extraction processes and the optimization of extraction variables for different sources of natural dyes can ensure the consistency of dye yields and quality, in turn limiting the cost of extraction and dyeing.

Anthocyanins are widely used as natural dyes because they are abundant and widespread in the plant kingdom, conferring pink, red, purple, violet and blue coloring in fruits, vegetables, and flowers. They are flavonoids with a C15 anthocyanidin skeleton consisting of a chromane ring joined to a second aromatic ring B at position 2 and are glycosylated and/or acylated on specific hydroxyl groups [6]. The stability of anthocyanins is dependent on pH, light, temperature, and intrinsic structure [7]. Ten different anthocyanins were identified in the leaves of radicchio, a red variety of chicory (*Cichorium intybus*), together with 64 other compounds including 31 flavonols and two flavone glucosides [8]. The anthocyanins previously identified in radicchio (Figure 1) were cyanidin-3-*O*-glucoside and pelargonidin-3-*O*-glucoside, with yields of 20 and 4.4 mg/100 g, respectively [9], as well as delphinidin-3-*O*-(6-malonyl)-glucoside, cyanidin-3-*O*-(6-malonyl)-glucoside, cyanidin-3-*O*-rutinoside, peonidin-3-*O*-glucoside, pelargonidin, and malvidin, with the yields not reported [10,11]. The quantity of secondary metabolites in red chicory strongly depends on the soil composition and growing conditions [11,12] as well as the precise variety. *C. intybus* comes in many varieties with different commercial uses, thus hampering botanical classification [10].

The general goal of this work is to develop a sustainable extraction method for red chicory anthocyanins as a green alternative source of natural dyes. In this framework, we used statistical experimental designs to develop a sustainable extraction protocol for red chicory leaves, followed by the analysis of the extracted anthocyanins and their utilization for textile dyeing.

## 2. Results

### 2.1. Definition of an Extraction Protocol for Red Chicory

#### 2.1.1. Gold-Standard Extraction

The solvent most widely used for the extraction of polyphenols is 1% HCl in methanol [13] and extraction is carried out at 4 °C to prevent photo-oxidation [14]. We, therefore, started from this basis and tested different conditions to optimize the protocol for red chicory (Figure 2).

We focused on the ratio of leaf fresh weight (LFW) to solvent volume (g/mL) and the use of sonication and different incubation times before centrifugation. We tested six different LFW/solvent ratios ranging from 1:3 to 1:60 (Figure 2a) and found that ratios of 1:20 or higher showed significantly better performance than lower ratios in terms of anthocyanin recovery. We selected the 1:20 ratio for subsequent experiments because of less solvent consumption. We observed no significant differences between the various solubilization steps, but the 30-min incubation was associated with the lowest variance and was therefore selected for subsequent experiments (Figure 2b). The final anthocyanin yield from red chicory was 1.01 ± 0.03 mg/g LFW and the procedure achieved the highest yields (mixing the red chicory powder with 1% HCl in methanol at a 1:20 ratio for 30 min at 4 °C) is defined herein as the gold-standard protocol.

#### 2.1.2. Establishment of the Sustainable Extraction Protocol

We tested a range of extraction solutions to determine the most efficient and environmentally sustainable method for the extraction of anthocyanins from red chicory. We initially compared water and 50% ethanol, each containing 1% HCl (Figure 3), and we extracted the red chicory powder for 30 min (as previously selected) but tested different incubation temperatures (4, 24, and 37 °C). This demonstrated that the different solvents and incubation temperatures had no significant impact on anthocyanin yields (two-way ANOVA, *p* > 0.05) but extraction at 37 °C resulted in a higher degree of variability, and this condition was therefore excluded. Removal of the acidic component (1% HCl) caused a major loss of extraction capacity (data not shown).

We then tested the impact of two different extraction temperatures (4 and 24 °C), four different concentrations of ethanol (0%, 12.5%, 25%, and 50%), and four different types of organic acid (acetic, citric, ascorbic and tartaric) at two different concentrations (0.2% and 2.0%) on anthocyanin yields after a 30-min incubation. This was achieved by applying a design-of-experiments approach. The multi-factor categorical design compared the four categorical factors (temperature, percentage ethanol, type of acid, and percentage of acid) and generated a multilevel factorial design with runs at each combination of factors at different levels (Figure 4a–d). We achieved significantly higher yields by using 2.0% tartaric acid in 50% ethanol at 4 °C (two-way ANOVA, *p* < 0.0001).

Having added the organic acids, we measured the final pH of each solution. This revealed (as expected) lower pH values for solutions containing tartaric or citric acid (Table 1). This higher acidification power reflects the lower dissociation constants of tartaric acid (pK_a_ = 2.89) and citric acid (pK_a_ = 3.13) for the first dissociation at 25 °C compared to ascorbic acid (pK_a_ = 4.17) and acetic acid (pK_a_ = 4.76).

We plotted a graphical representation of the estimated change in yield under different conditions for each of the four categorical factors (temperature, percentage ethanol, type of acid, percentage of acid). The main effects plot indicates that the major variables affecting anthocyanin yields are the type and concentration of acid, whereas the temperature has only a minor impact and the ethanol concentration does not affect yields in a linear manner beyond a concentration of 25% (Figure 5). Further experiments demonstrated that higher concentrations of tartaric acid did not increase the yields any further (data not shown).

### 2.2. Stability of Extracted Anthocyanins over Time

We used protocols that yielded the highest anthocyanin concentrations (four different ethanol concentrations combined with 2% tartaric acid at 4 °C) and then measured the stability of the anthocyanins in the pure extract, a three-fold concentrate and a lyophilized extract at two different temperatures (4 and 24 °C) over eight-time points (1 h, 24 h, 48 h, 1 week, 2 weeks, 1 month, 3 months and 6 months). We compared the anthocyanin content of each stored sample to that of a matching sample that was analyzed immediately after extraction. We found that the form of the stored extract was critical, strongly affecting the susceptibility of the anthocyanins to degradation in a manner depending mainly on the storage temperature and the percentage of ethanol in the original extract (Figure 6, Figure 7 and Figure 8). The anthocyanins in the pure extract were remarkably susceptible to degradation at 23 °C and in the presence of higher concentrations of ethanol at both temperatures, with 100% loss after 3 months at 23 °C in the presence of 50% ethanol (Figure 6). The three-fold concentrate showed a similar trend at 23 °C (although the anthocyanins were slightly less sensitive to degradation compared to the pure extract) but reducing the temperature to 4 °C significantly improved anthocyanin stability over time (Figure 7). The lyophilized samples were the most stable. There was no significant temperature-dependent effect on samples extracted with water or 25–50% ethanol until 1 month of storage (*t*-test, *p* > 0.05; Figure 8a,c,d) but we observed a significant temperature-dependent effect after 6 months in the samples extracted with ethanol (*t*-test, *p* < 0.05; Figure 8b–d). Extraction in water resulted in better anthocyanin stability over time, with no significant loss of anthocyanin after the storage of lyophilized samples at 4 or 23 °C.

### 2.3. Molecular Characterization of the Extracted Dyes

Red chicory leaf extracts were analyzed by HPLC-DAD to precisely quantify the content of the single anthocyanin species, which were structurally elucidated and identified through UPLC-MS. Samples prepared using water acidified with 2% (*v*/*v*) tartaric acid and different percentages of ethanol (0%, 12.5%, 25%, and 50% *v*/*v*) were analyzed by HPLC-DAD immediately after extraction, with the gold-standard protocol used as a reference. The chromatograms showed three main peaks at 520 nm (Appendix A), whereas the reference sample featured an additional peak (Appendix A) suggesting the presence of four putative anthocyanins in red chicory. We also compared the anthocyanin profiles of lyophilized extracts prepared with the same four solvents and stored at 4 or 23 °C for 6 months. HPLC-DAD analysis revealed the presence of two further peaks not present in the fresh extracts (Appendix A). All the extracts described above were then analyzed by UPLC-MS to identify the specific anthocyanins (Table 2).

UPLC-MS analysis showed that the water and ethanol extracts mainly contained cyanidin and delphinidin derivatives, as indicated by the presence of fragments with *m*/*z* values of 287.056 (cyanidin aglycone) and 303.051 (delphinidin aglycone) for anthocyanins in peaks 1, 2, 3, 5, and 6 (Appendix A). Moreover, red chicory also appears to conjugate anthocyanins with malonyl (+86 Da) and acetyl (+42 Da) residues. Peak 4 (Appendix A) was only detected in methanol extracts and included an anthocyanin with *m*/*z* values of 549.1238 [M^+^] in positive ionization mode and 547.1088 [M-2H]^–^ in negative ionization mode. The fragmentation of this parent ion yielded cyanidin aglycone products in both positive mode (*m*/*z* = 287.056) and negative mode (*m*/*z* = 284.033). Moreover, a fragment of *m*/*z* = 447.0927 detected in negative mode suggested the neutral loss of 100 Da from a cyanidin with a sugar moiety. This last value is usually associated with a loss of a succinyl residue, suggesting the fragment corresponds to a cyanidin succinyl hexoside. Peak 5 appeared only in lyophilized samples stored at 23 °C, and UPLC-MS analysis in positive mode revealed the presence of a parent ion (*m*/*z* = 643.1303) that fragmented to yield a product of *m*/*z* = 287.056, suggesting a cyanidin derivative (cyanidin derivative 1 in Table 2). Peak 6 appeared in lyophilized ethanol extracts stored at either 4 or 23 °C and might include a cyanidin derivative (cyanidin derivative 2 in Table 2) as suggested by the fragmentation of the parent ion (*m*/*z* = 563.1401) to yield an aglycone product ion (*m*/*z* = 287.056).

All red chicory extracts freshly prepared at 4 °C showed similar anthocyanin profiles (Figure 9, charts in left column). Cyanidin 3-*O*-(6″-*O*-malonyl)-glucoside and cyanidin 3-*O*-(6″-*O*-acetyl)-glucoside were the predominant anthocyanins (~78% of the total signal at 520 nm), followed by cyanidin 3-*O*-glucoside (~11% of the total signal at 520 nm) and the sum of cyanidin 3,5-di-*O*-(6″-*O*-malonyl)-glucoside, delphinidin 3-*O*-(6″-*O*-malonyl)-glucoside and delphinidin 3-*O*-(6″-*O*-acetyl)-glucoside (~12% of the total signal at 520 nm). Cyanidin succinyl hexoside was the least abundant anthocyanin species (~4% of the total signal at 520 nm) and was only detected in the methanol extracts.

The lyophilized samples stored for 6 months showed specific anthocyanin profiles depending on the extraction solvent and storage temperature (Figure 9, charts in middle and right columns). The presence of ethanol in the solvent resulted in the appearance of a cyanidin derivative that was not present when red chicory was extracted in water (cyanidin derivative 2, peak 6). This component represented ~7% and ~10% of the total signal at 520 nm in samples stored at 4 and 23 °C, respectively. Storage at 23 °C significantly reduced the content of cyanidin 3-*O*-(6″-*O*-malonyl)-glucoside plus cyanidin 3-*O*-(6″-*O*-acetyl)-glucoside from ~73% to ~40% of the total signal at 520 nm. In contrast, the level of cyanidin 3-*O*-glucoside increased from ~12% to ~37% of the total signal at 520 nm, whereas the combined content of cyanidin 3,5-di-*O*-(6″-*O*-malonyl)-glucoside, delphinidin 3-*O*-(6″-*O*-malonyl)-glucoside and delphinidin 3-*O*-(6″-*O*-acetyl)-glucoside increased from ~10% to ~13% of the total signal at 520 nm. Cyanidin derivative 1 (peak 5) was detected solely in lyophilized samples stored at 23 °C but accounted for only ~3% of the total signal at 520 nm.

### 2.4. Dyeing Properties of Extracted Anthocyanins

Anthocyanins extracted in water acidified with 2% (*v*/*v*) tartaric acid were used to dye wool yarns to evaluate their dyeing properties in the presence and absence of potassium alum as a mordant (to increase color fastness and intensity). The two processes (WoolP_1 and WoolP_2) are compared in Figure 10.

The woolen yarns were easily dyed with the extracted anthocyanins in the absence of the mordant (Figure 11b), but the presence of potassium alum achieved a brighter and more intense coloration (Figure 11a). We presume that aluminum can form weak coordination complexes with the dye molecules, resulting in more vivid color. The adsorption of the dye was quantified by using UV/vis spectrophotometry to determine the amount of dye remaining in the bath. We found that more dye was adsorbed in the absence of mordant (98%) compared to the yarn treated with potassium alum (88%). The kinetic behavior of the WoolP_1 and WoolP_2 dyeing processes was investigated in detail by calculating pseudo-first-order and pseudo-second-order models (Table 3 and Figure 12). The kinetic behavior of both processes was similar (comparable k values). The pseudo-second-order model (Figure 12b,d) was the best fit for both processes, with a correlation coefficient of R^2^ ≥ 0.990. These results suggest the pseudo-second-order kinetic model closely describes the adsorption of anthocyanins by wool samples in the presence and absence of potassium alum.

The pH-dependent color-changing properties of anthocyanins are well-known. As anticipated, the pink-dyed yarns from processes WoolP_1 and WoolP_2 turned green (WoolG_1 and WoolG_2, respectively) after washing with a European Colorfastness Establishment (ECE) reference textile detergent (Figure 13). However, they returned to pink again after exposure to mild acid. The wool dyed in the presence of the mordant turned a more intense green (WoolG_1) compared to wool dyed without mordant, which appeared to be discolored (WoolG_2). Accordingly, different colors can easily be obtained by varying the dyeing conditions (mordant and pH) and can therefore be exploited for applications in the fashion industry.

### 2.5. Colour Fastness

Finally, we characterized the four samples for washing fastness, acid and alkaline perspiration fastness, and light fastness (Table 4). As anticipated based on the color-change analysis, WoolP_1 and WoolP_2 turned green during the washing fastness test with ECE soap (pH~8) and it was not possible to properly evaluate the extent of fading. Indeed, the assigned low color fastness values (Table 4) did not correspond to low color intensity after washing because we generally observed bright and intense colors, albeit of a different hue. The low values observed for light fastness may be due to the organic nature of the dye. The acid and alkaline perspiration tests also resulted in low values for all samples. The assessment of the extent of staining compared to the adjacent multifiber strip in the washing and perspiration tests is also shown in Table 4. The anthocyanin dye achieved very good fastness properties when the multifiber strip was examined, with only minimal staining of the cotton witness. 

## 3. Discussion

Agri-food industry waste streams offer a valuable resource-rich in bioactive compounds (phenols, peptides, carotenoids, anthocyanins, and fatty acids), fibers, and enzymes that can be used to produce functional foods, drugs, cosmetics, and dyes. The circular economy is an efficient option in the medium to long term to re-introduce such byproducts as raw materials into the value chain, using sustainable technology for their extraction [15]. Accordingly, we have developed a scalable and environmentally benign protocol for the extraction of anthocyanins from red chicory based on the use of non-toxic extraction solvents (mixtures of water, ethanol, and organic acids). This is a sustainable process for the production of safe and biodegradable dyes from a renewable waste stream, using a protocol designed to minimize environmental impact. We carried out small-scale studies using a design-of-experiments approach to optimize extraction efficiency and then evaluated the resulting anthocyanin profiles.

The total anthocyanin content extracted from red chicory leaves was unaffected by the proportion of ethanol in the aqueous solvent or by the extraction temperature when a strong acid was included (2% HCl). However, when testing a range of natural organic acids, these parameters had a significant effect on the final yield of anthocyanins. Ascorbic and acetic acid were less efficient than citric and tartaric acid and had less impact on the variability caused by other parameters (temperature and ethanol content). Citric acid was more efficient and had a moderate impact on the other parameters, but 2% tartaric acid was the most efficient overall and maximized the variability in yields caused by the temperature and ethanol concentration in the extraction buffer. Accordingly, we were able to define an optimal protocol based on the extraction of red chicory powder at 4 °C for 30 min using 50% ethanol containing 2% tartaric acid as the solvent, matching the efficiency of the gold-standard protocol based on methanol acidified with 2% HCl under the same conditions (no significant difference observed in a *t*-test, *p* > 0.05).

We characterized the extracts by evaluating their stability over time when stored as pure extracts, three-fold concentrates, or lyophilized powders at two different temperatures (4 and 23 °C). We found that the lyophilization of aqueous extracts (extraction buffer = 2% tartaric acid in water with no ethanol) followed by storage at 4 °C preserved the anthocyanin contents for 6 months, whereas the storage of pure extracts or three-fold concentrates revealed a strong negative effect on anthocyanin stability caused by the higher storage temperature and by the presence of ethanol in the extraction buffer. By lowering the water activity of the matrix through the sublimation of water molecules at low temperatures, lyophilization reduces the reactivity of anthocyanins, including their conversion to colorless hemiketal and chalcone forms that occur naturally in aqueous environments [16]. This freeze-drying method has already been used successfully by others to preserve the anthocyanin content of other plant matrices for 6 months, including extracts of sweet cherry [17] and elderberry [18]. Therefore, although the most efficient extraction process required a solvent containing 50% ethanol, the presence of ethanol limits the post-extraction stability of anthocyanins over time when stored as pure extracts, concentrates, or lyophilized powder. The degradation kinetics of anthocyanins in the presence of increasing concentrations of ethanol have been associated with the disruption of π-interactions between the aromatic rings [19]. In an aqueous solution, these interactions stack the planar structures of anthocyanins (a phenomenon known as self-association), shielding their cores from nucleophilic attacks that can lead to hydrolysis or oxidation. Ethanol is thought to interfere with this stacking phenomenon to indirectly cause irreversible degradation of the chromophores, triggering the color loss we observed in the pure extracts and concentrates containing 50% ethanol. When using water containing 2% tartaric acid, the temperature-dependent degradation of anthocyanins was ameliorated, especially when stored as a lyophilized powder (multiple *t*-tests, *p* > 0.05). We, therefore, selected storage at 23 °C in our optimized sustainable protocol.

The total anthocyanin content of red chicory leaf extracts prepared using our optimized sustainable protocol (70.1 ± 1.8 mg/100 g LFW) was higher than previously reported. For example, Lavelli [11] achieved maximum yields of 65.3 mg/100 g LFW by extraction with 50% methanol containing 4% formic acid at room temperature, whereas Migliorini et al. [9] achieved maximum yields of 73.53 ± 0.13 mg/100 g LFW by extraction with water acidified with acetic acid (pH 2.5 at 62.4 °C).

Red chicory leaves have previously been shown to accumulate various anthocyanins, particularly cyanidin-3-*O*-galactoside, cyanidin-3-*O*-glucoside, cyanidin-3-*O*-(6-malonyl)-glucoside, cyanidin-3-*O*-rutinoside, cyanidin-3,5-di-*O*-(6-*O*-malonyl)-glucoside, cyanidin-3-*O*-(-*O*-acetyl)-glucoside, and cyanidin-3-*O*-glucuronyl-5-*O*-hexoside [8,20]. However, delphinidin 3-*O*-(6-malonyl)-glucoside, delphinidin-3-*O*-(6-*O*-malonyl)-glucoside-5-*O*-glucoside, pelargonidin-3-*O*-glucoside, peonidin-3-*O*-glucoside, and malvidin-3-*O*-glucoside were also present in smaller amounts [8]. We used UPLC-MS to characterize red chicory extracted in water acidified with 2% (*v*/*v*) tartaric acid and containing 0%, 12.5%, 25% or 50% (*v*/*v*) ethanol, revealing that cyanidin 3-*O*-(6″-*O*-malonyl)-glucoside and cyanidin 3-*O*-(6″-*O*-acetyl)-glucoside were by far the most abundant anthocyanins. The former was previously reported as the most abundant anthocyanin in red chicory [21] but the latter might be interesting for commercial purposes because acylated cyanidin derivatives were more stable than their non-acylated counterparts during the fermentation of blackberry wine and juice, and were also resistant to degradation by commercial pectinase enzyme preparations [22]. Interestingly, we identified cyanidin succinyl hexoside in acidified methanol extracts, which (to the best of our knowledge) have not been reported before.

Lyophilized extracts stored for 6 months featured two additional peaks at 520 nm that were identified as two different cyanidin derivatives. These were not present in the fresh extracts, suggesting that a conversion reaction takes place during lyophilization and/or storage. The high levels of cyanidin-3-*O*-glucoside in lyophilized samples stored at 23 °C suggest that a degradation process might occur at the expense of more complex anthocyanins, resulting in the accumulation of the simplest glycosylated form.

Anthocyanins undergo diverse reactions in a pH-dependent manner [23]. Here we evaluated their behavior as textile dyes and found that wool was easily dyed both in the presence and absence of potassium alum as a mordant. However, the presence of potassium alum conferred a brighter and more intense color, presumably reflecting the ability of aluminum to form complexes with the dye. Both processes (with and without the mordant) fitted best to pseudo-second-order kinetics (R^2^ ≥ 0.990) and followed similar curves. Anthocyanins are known to change color when exposed to pH variations and accordingly, the pink-dyed yarns turned green when washed with a slightly alkaline reference detergent and changed back to pink on exposure to mild acid. This property can be exploited for the development of garments and textiles over a range of colors. All samples (pink and green) showed similar good results for washing fastness (the low values recorded in Table 4 reflect the color change rather than fading) and only minimal staining was observed on the cotton witness strip. The green yarn samples (Wool_G1 and WoolG_2) showed good values for basic perspiration fastness, whereas the pink samples (WoolP_1 and WoolP_2) faded and changed color. All samples (pink and green) showed poor acid perspiration fastness due to fading, and poor light fastness probably due to the organic nature of the dyes. The use of co-pigmentation strategies could improve this behavior by stabilizing the anthocyanins.

The dyes described herein are anthocyanin-rich extracts isolated from renewable botanical sources (leaf waste) using innocuous solvents, offering a sustainable alternative to the popular synthetic dyes of today, which are derived from hazardous petrochemical sources using reactions that require toxic solvents and that generate toxic intermediates. The dyes are also biodegradable, offering the potential for a fully sustainable and environmentally benign textile-dyeing process.

## 4. Materials and Methods

### 4.1. Plant Material

Mature leaves of red chicory (*C. intybus* var. *silvestre* “Verona”) were washed thoroughly and ground to a fine powder under liquid nitrogen. The ground powder was packed in plastic bags and stored at −80 °C.

### 4.2. Gold-Standard Extraction of Polyphenols from Plant Material

Anthocyanins were extracted from plant material as previously described [24] with minor modifications. The ground leaf powder was resuspended in five volumes of 99:1 methanol:HCl (*v*/*v*) and sonicated for 10 min at 4 °C. Methanol extracts were centrifuged at 20,000× *g* for 10 min to remove cell debris. The supernatant was collected and stored at −20 °C.

### 4.3. Quantification of Anthocyanins

The total anthocyanin content of extracts was determined by UV/vis spectrophotometry using a Genequant 1300 device (Biochrom, Cambourne, UK) to record the absorption spectrum at 520 nm. The anthocyanin content was calculated using Equation (1):(1)$$Total\;anthocyanins\;\left(\frac{mg}{g}LFW\right)=\frac{Abs\times DF\times SR}{\varepsilon \times 1000}$$
where ε is the molar extinction coefficient for cyanidin-3-glucoside (ε  =  26,900 M^−1^ cm^−1^), *Abs* is the absorbance, *DF* is the dilution factor and *SR* is the ratio of *LFW*/solution.

### 4.4. Design-of-Experiments

Statistical experimental designs were evaluated using Statgraphics CENTURION XVIII software. A multi-factor categorical design was used to compare levels of four categorical factors: temperature (4 and 23 °C), percentage ethanol (0%, 12.5%, 25% and 50% *v*/*v*), type of acid (acetic, ascorbic, citric, and tartaric acid), and percentage of acid (0.2 and 2% *w*/*v*). The procedure created a multilevel factorial design with runs for each combination of the levels representing each factor. The anthocyanin content for each condition was inputted back into the software. Multifactor ANOVA was used to determine the significance of the factors and their interactions.

### 4.5. Stability of Extracted Anthocyanins 

Anthocyanin stability was tested at 4 and 24 °C. We extracted 25 g of frozen red chicory leaf powder with each buffer and divided the extracts into 1-mL aliquots for direct analysis (pure extracts), concentration in a SpeedVac system (Thermo Fischer Scientific, Waltham, MA, USA) or lyophilization in a Benchtop freeze dryer (Christ, Osterode am Harz, DE). The lyophilized material was resuspended in one volume of water before reading absorbance values. Anthocyanin stability was tested at eight time points (1 h, 24 h, 48 h, 1 week, 2 weeks, 1 month, 3 months, and 6 months). Three sample replicates were tested for each condition and time point. The total anthocyanin content in stored samples was compared to that of a freshly prepared extract-treated under the same conditions. In the lyophilization experiment, the total anthocyanin content was also determined in the extract just after lyophilization but before storage (T0).

### 4.6. LC-MS Analysis and Data Processing

Fresh or stored samples were vortexed for 30 s then centrifuged at 13,000× *g* for 10 min at 4 °C. The extracts were diluted 1:2 (*v*/*v*) with LC-MS grade water (Honeywell), passed through Minisart RC4 filters with 0.2-µm pores (Sartorius) and 30 µL was injected into the HPLC-DAD device (Beckman Coulter, Brea, USA). The chromatographic column was an Alltima C18 (150 × 2.1 mm; 3 µm of particle size; Allthech) equipped with a guard column (7 × 2.1 mm; Allthech). The flow rate set at 0.2 mL/min and solvents were: water plus 5% (*v*/*v*) acetonitrile and acidified with 0.5% (*v*/*v*) formic acid (A) and acetonitrile (B). The gradient was 0–10% B in 2 min, 10–20% B in 10 min, 20–25% B in 2 min, 25–70% B in 7 min, isocratic elution at 70% B for 5 min, 70–90% B in 1 min, isocratic elution at 90% B for 4 min, 90% B–0% B in 1 min, and a final equilibrium of 18 min in 0% B. HPLC-DAD instrument details and analysis parameters were described in previous work [25]. For metabolomics analysis, the extracts were diluted 1:10 (*v*/*v*) with LC-MS grade water, passed through Minisart RC4 filters as above, and 2 µL was injected into the UPLC-MS system (Waters, Milford, USA). The chromatographic column was an Acquity BEH C18 (100 × 2.1 mm; 1.7 µm of particle size; Waters), the flow rate set at 0.35 mL/min, and solvents were: water acidified with 0.1% (*v*/*v*) formic acid (A) and acetonitrile (B). The gradient was set as follows: 0–1 min, 1% B; 1–10 min, 1–40% B; 10–13.50 min, 40–70% B; 13.50–14.00 min, 70–99% B; 14.00–16.00 min, 99% B; 16.00–16.10 min, 99–1% B (initial conditions). UPLC-MS instrument details and analysis parameters were described in previous work [26].

### 4.7. Wool Dyeing with Anthocyanins

The wool yarns were dyed with 50% o.w.f. (on weight fibers) anthocyanin, 10% o.w.f. sodium sulfate and 1% o.w.f. sodium dodecylsulfate with a yarn-to-liquor ratio of 1:40 in an AHIBA Nuance Top Speed II device shaking at 35 rpm. The baths were heated to 95 °C at a rate of 1.3 °C/min and then held for 1 h, with an additional 20% o.w.f. sodium sulfate added after 30 min. The dyed materials were left in dye solution at room temperature for 12 h and then rinsed thoroughly with tap water, squeezed, and dried in an oven at 50 °C. Dyeing was carried out with and without the addition of 20% o.w.f. potassium alum at the beginning of the process.

### 4.8. Evaluation of Dyeing Process Kinetics and Textile Fastness

#### 4.8.1. Dyeing Kinetic Parameters

The quantity of dye adsorbed by the wool fibers was evaluated by measuring the dye remaining in the bath. Absorbance at 520 nm was measured using a double-beam Perkin Elmer 200 UV/vis spectrophotometer (Perkin Elmer, Waltham, MA, USA) and the amount of dye was determined against a calibration curve. The amount of dye adsorbed onto the fibers was then calculated using Equation (2):
q_t_ = (C_0_ − C_t_) (V/m)(2)
where q_t_ is the quantity of dye adsorbed at equilibrium and time t (mg/g), C_0_ is the initial dye concentration, C_t_ is the dye concentration at time t, V is the volume of the dyeing bath, and m is the mass of the wool yarn. Pseudo-first-order and pseudo-second-order models were used to evaluate the dye adsorption behavior. The pseudo-first-order equation is generally suitable for the initial stage of the adsorption process, whereas the pseudo-second-order equation predicts the adsorption behavior during the whole process. The kinetics of adsorption was determined by analyzing the dye adsorbed from an aqueous solution over the time 0–12 h. The pseudo-first-order rate is shown in Equation (3):
ln(q_e_ − q_t_) = lnQ_e_ − k_1t_(3)
where q_t_ is the quantity of dye adsorbed at equilibrium and time t (mg/g), q_e_ is the theoretical equilibrium adsorption (mg/g), and k_1_ is the pseudo-first-order absorption rate constant (/min).

The pseudo second-order kinetic rate is shown in Equation (4):t/q_t_ = 1/k_2_(q_e_)2 + t/q_e_(4)
where q_t_ is the quantity of dye adsorbed at the time t, q_e_ is the theoretical equilibrium adsorption (mg/g), and k_2_ is the pseudo second-order adsorption rate constant (/min).

#### 4.8.2. Washing Fastness

Washing fastness describes resistance to fading and staining when a dyed fabric is washed. The washing fastness of the dyed yarns was evaluated according to ISO 105-C06 using 4 g/L ECE detergent (Swissatest Testmaterialien) for 30 min at 40 °C (pH~8). The samples were attached to multifiber fabric and cotton fabric witnesses by sewing along all the sides to evaluate the tendency to stain other materials. After the wash, each sample was opened and dried at room temperature. Washing fastness scores for fading and staining range from 1 to 5, from the poorest to the best performance.

#### 4.8.3. Acid and Alkaline Perspiration Fastness

Acid and alkaline perspiration fastness describe resistance to fading when a dyed fabric comes in contact with sweat, which is tested using acid and alkaline artificial perspiration mixtures prepared according to AATCC Test Method 15-1994. The alkaline artificial perspiration mixture was prepared by dissolving 0.5 g l-histidine-HCl.H_2_O, 5.0 g NaCl and 5 g HNa_2_PO_4_ in 1 L distilled water and adjusting to pH 8 with 0.1 M NaOH. The acid artificial perspiration mixture was prepared by dissolving 0.5 g l-histidine-HCl.H_2_O, 5.0 g NaCl, and 2.2 g HNa_2_PO_4_ in 1 L of distilled water and adjusting to pH 5.5 with 0.1 M HCl. The samples were attached to the multifiber fabric and cotton fabric witnesses as above and were placed in a dish containing 100 mL of the perspiration solution for 30 min. The samples were then pressed between two glass plates and placed in an upright position in a perspirometer (Atlas Electric Devices C.C., Chigaco, IL, USA) at 12.5 KPa for 4 h at 37 °C. Each sample was opened and dried at room temperature. Perspiration fastness scores for fading and staining range from 1 to 5, from the poorest to the best performance.

#### 4.8.4. Light Fastness

Light fastness was evaluated using an Atlas Xenotest Alpha (Alpha Test, Atlas, Chicago, IL, USA) device equipped with a xenon lamp according to ISO 105. The dyed yarns were exposed to artificial light under controlled conditions with a set of reference materials. The color fastness was assessed by comparing the change in color of the test sample with that of the reference strips. The lightfastness score ranges from 1 to 7, from the poorest to the best performance.

### 4.9. Statistical Analysis

All assays were carried out at least three times and the data are presented as means ± standard deviations. Statistical significance in the gold-standard extraction experiments was established by one-way analysis of variance (ANOVA) followed by Tukey’s test for multiple comparisons, with a threshold of *p* < 0.05. For the new sustainable protocol, we evaluated the effect of different solvents on anthocyanin yields by two-way ANOVA followed by Šidák’s test for multiple comparisons, with a threshold of *p* < 0.05. Statistical significance in the anthocyanin stability experiments was established by multiple t-tests followed by Holm–Šidák correction for multiple comparisons, with a threshold of *p* < 0.05.

## 5. Conclusions

The use of agri-food waste streams as a source of natural dyes aligns with the concept of a green circular economy, converting waste into valuable resources. In this sustainable context, we set up an environmentally benign method, supported by robust statistical analysis, for the extraction of anthocyanins from red chicory leaves. The anthocyanins extracted by our optimized sustainable protocol showed high stability over time when lyophilized and had good textile dyeing properties. Such innovations will help to reduce waste in the agri-food industry as well as support sustainable and circular manufacturing.

## Figures and Tables

**Figure 1 molecules-26-06775-f001:**
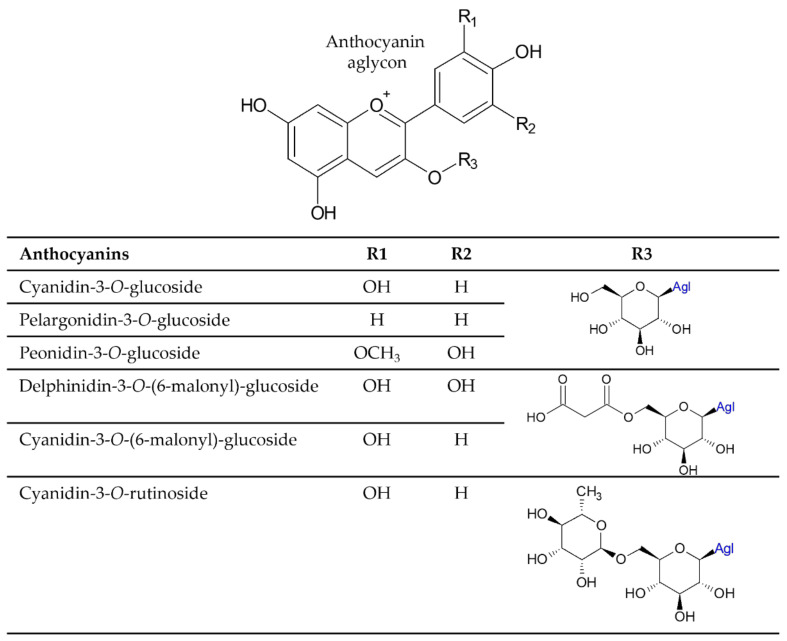
Examples of anthocyanins structures are reported in red chicory. Agl: anthocyanin aglycon.

**Figure 2 molecules-26-06775-f002:**
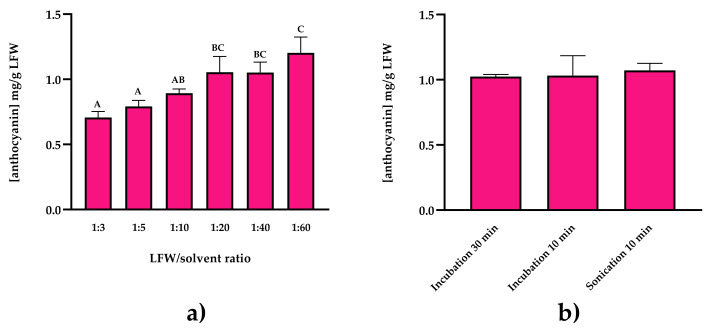
Anthocyanin yields were obtained with (**a**) different ratios of leaf fresh weight (LFW) to solvent volume (g/mL) and (**b**) different solubilization methods. Data are means ± SD (*n* = 3 independent experiments; one-way ANOVA with Tukey’s post hoc test, *p* < 0.05, significant differences are indicated by different letters).

**Figure 3 molecules-26-06775-f003:**
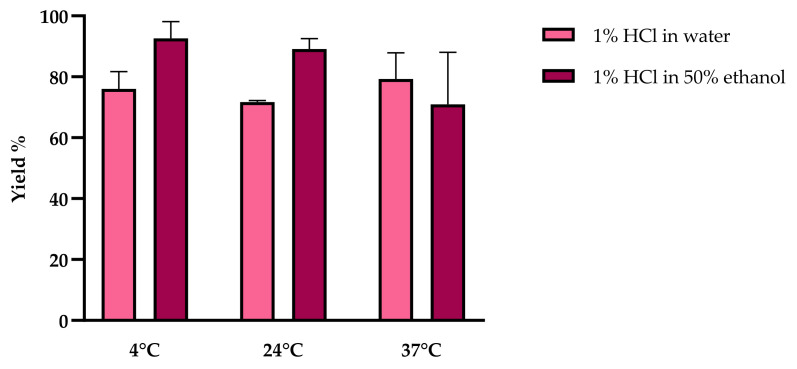
Anthocyanin yields were obtained by extraction with different solvents (water or 50% ethanol, each containing 1% HCl) at different incubation temperatures (4, 24, and 37 °C). Values are percentages (yield %) relative to the gold-standard method. Data are means ± SD (*n* = 3 independent experiments; two-way ANOVA with Šidák’s post hoc test, *p* < 0.05).

**Figure 4 molecules-26-06775-f004:**
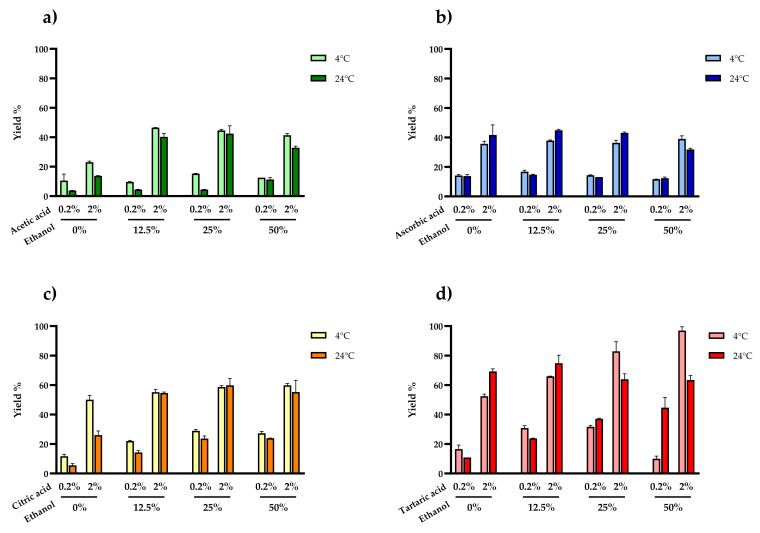
Anthocyanin yields were obtained at two different temperatures (4 or 24 °C), with four concentrations of ethanol (0%, 12.5%, 25%, and 50%) and four different types of organic acid (acetic (**a**), ascorbic (**b**), citric (**c**) and tartaric (**d**)) at two different concentrations (0.2% and 2.0%). Values are percentages (yield %) relative to the gold-standard method. Data are means ± SD (two replicates at each combination of factor levels, randomization of the run order).

**Figure 5 molecules-26-06775-f005:**
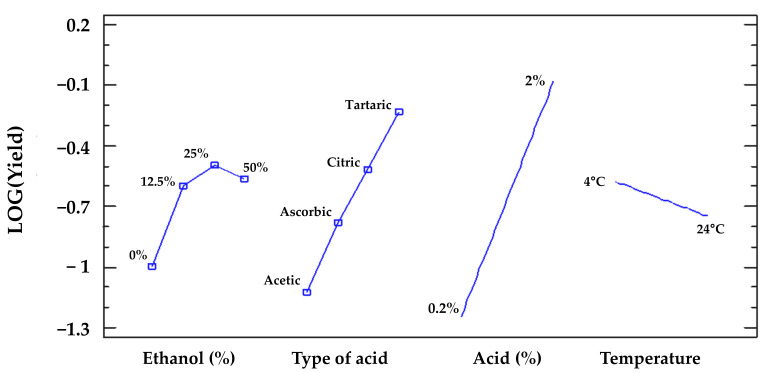
The main effects plot shows changes in response when each of the four categorical factors (temperature, percentage ethanol, type of acid, percentage of acid) is moved from its lowest to its highest level.

**Figure 6 molecules-26-06775-f006:**
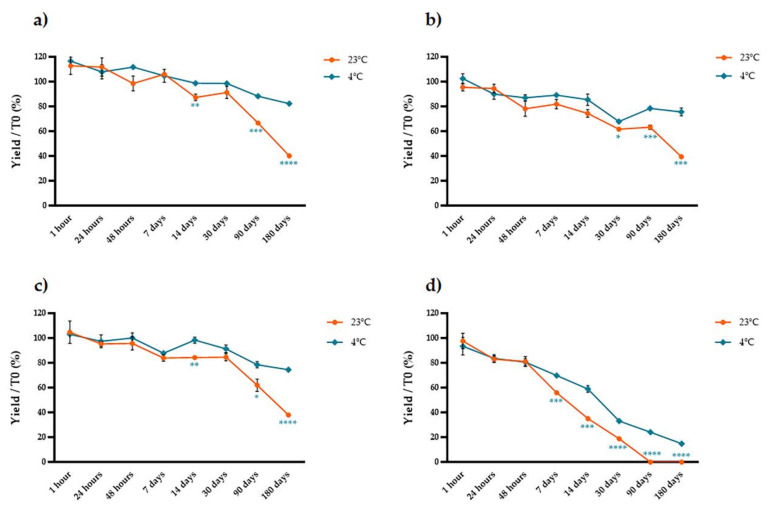
Stability over time of pure extracts prepared with different solvents and stored at 23 °C (blue line) or 4 °C (red line). The samples were extracted with (**a**) water, (**b**) 12.5% ethanol, (**c**) 25% ethanol, and (**d**) 50% ethanol. The y axis shows the yield as a percentage relative to the samples that were analyzed immediately after extraction. Significant differences between the two storage temperatures are indicated with asterisks (* *p* < 0.05, ** *p* < 0.01, *** *p* < 0.001, **** *p* < 0.0001). Data are means ± SD (*n* = 3 independent experiments; multiple *t*-tests with Holm–Šidák correction, *p* < 0.05).

**Figure 7 molecules-26-06775-f007:**
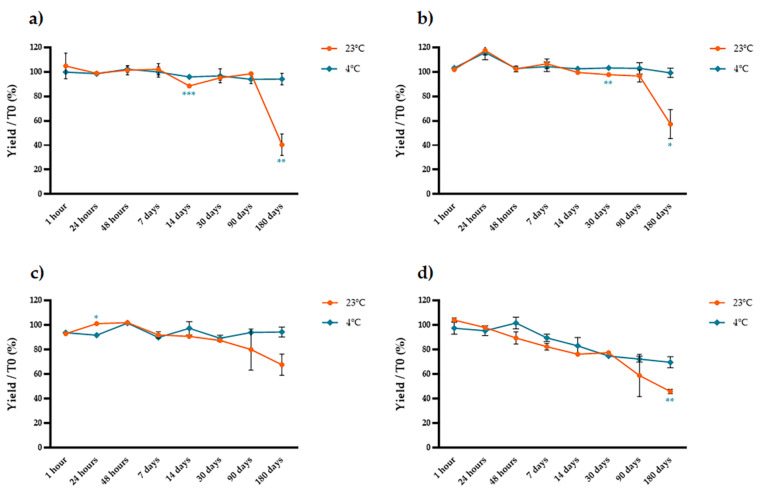
Stability over time of three-fold concentrated extracts prepared with different solvents and stored at 23 °C (blue line) or 4 °C (red line). The samples were extracted with (**a**) water, (**b**) 12.5% ethanol, (**c**) 25% ethanol, and (**d**) 50% ethanol before concentration. The y axis shows the yield as a percentage relative to the samples that were analyzed immediately after extraction. Significant differences between the two storage temperatures are indicated with asterisks (* *p* < 0.05, ** *p* < 0.01, *** *p* < 0.001, **** *p* < 0.0001). Data are means ± SD (*n* = 3 independent experiments; multiple *t*-tests with Holm–Šidák correction, *p* < 0.05).

**Figure 8 molecules-26-06775-f008:**
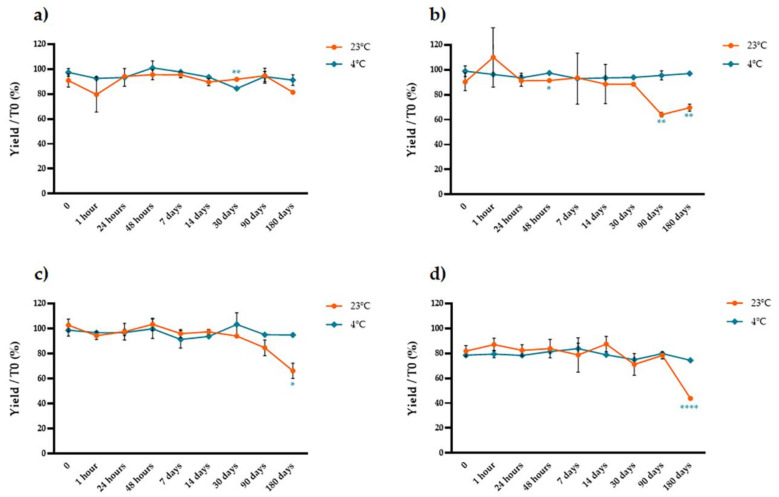
Stability over time of lyophilized extracts prepared with different solvents and stored at 23 °C (blue line) or 4 °C (red line). The samples were extracted with (**a**) water, (**b**) 12.5% ethanol, (**c**) 25% ethanol, and (**d**) 50% ethanol before lyophilization. The y axis shows the yield as a percentage relative to the samples that were analyzed immediately after extraction. Significant differences between the two storage temperatures are indicated with asterisks (* *p* < 0.05, ** *p* < 0.01, *** *p* < 0.001, **** *p* < 0.0001). Data are means ± SD (*n* = 3 independent experiments; multiple *t*-tests with Holm–Šidák correction, *p* < 0.05).

**Figure 9 molecules-26-06775-f009:**
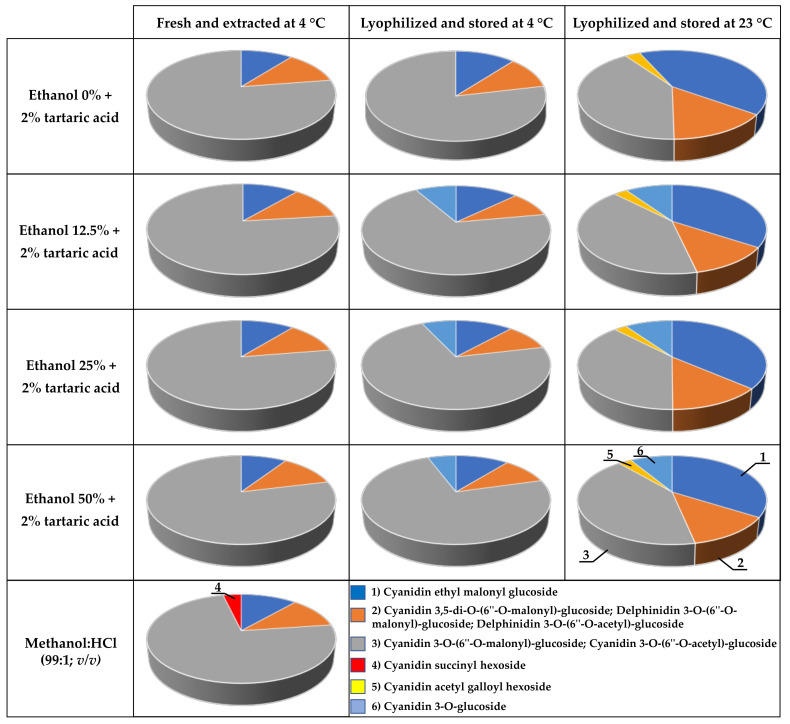
Anthocyanin profile of red chicory extracts. Values refer to peak areas in arbitrary units (AU) extrapolated from HPLC-DAD data at 520 nm. Cyan: relative amount of cyanidin 3-*O*-glucoside. Orange: sum of cyanidin 3,5-di-*O*-(6″-*O*-malonyl)-glucoside, delphinidin 3-*O*-(6″-*O*-malonyl)-glucoside and delphinidin 3-*O*-(6″-*O*-acetyl)-glucoside. Gray: sum of cyanidin 3-*O*-(6″-*O*-malonyl)-glucoside and cyanidin 3-*O*-(6″-*O*-acetyl)-glucoside. Red: relative amount of cyanidin succinyl hexoside. Yellow: relative amount of cyanidin derivative 1. Blue: relative amount of cyanidin derivative 2. Numbers indicate the corresponding peaks in Appendix A.

**Figure 10 molecules-26-06775-f010:**
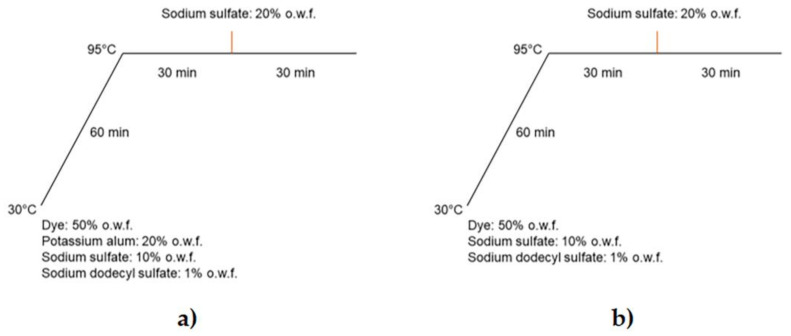
Protocols for the dyeing of woolen yarn with plant-derived anthocyanins at a load of 50% on weight fibers (o.w.f.) in the presence ((**a**), WoolP_1) or absence ((**b**), WoolP_2) of potassium alum as a mordant.

**Figure 11 molecules-26-06775-f011:**
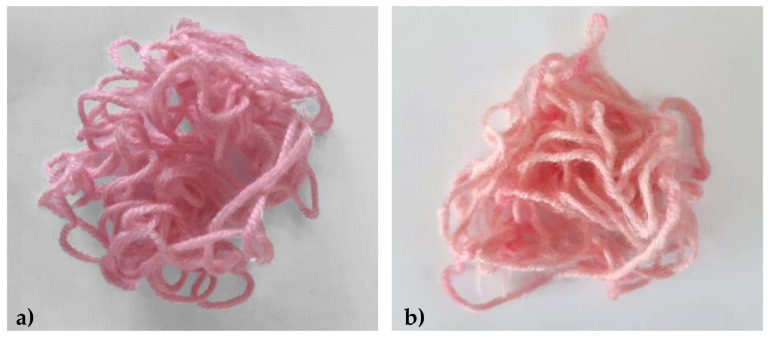
The visual appearance of wool yarn dyed with plant-derived anthocyanins in the presence ((**a**), WoolP_1) or absence ((**b**), WoolP_2) of potassium alum as a mordant.

**Figure 12 molecules-26-06775-f012:**
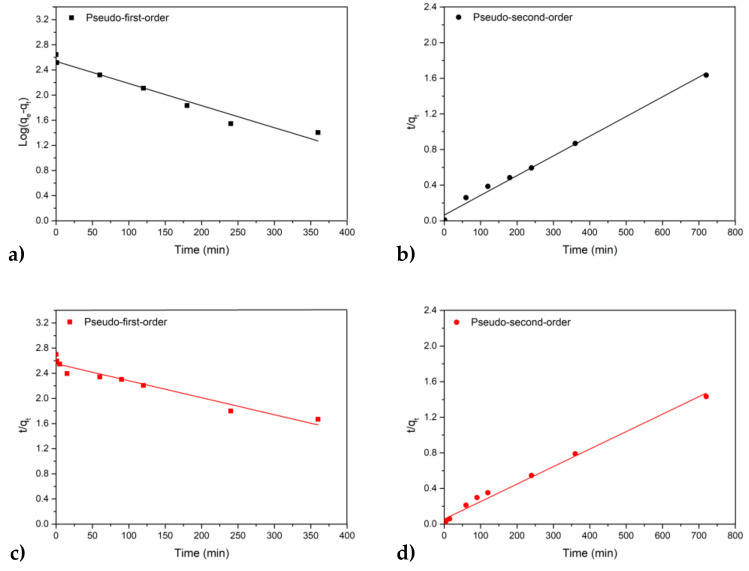
Kinetic parameters evaluated for the processes WoolP_1 (**a**,**b**) and WoolP_2 (**c**,**d**) using pseudo-first-order (**a**,**c**) and pseudo-second-order (**b**,**d**) models.

**Figure 13 molecules-26-06775-f013:**
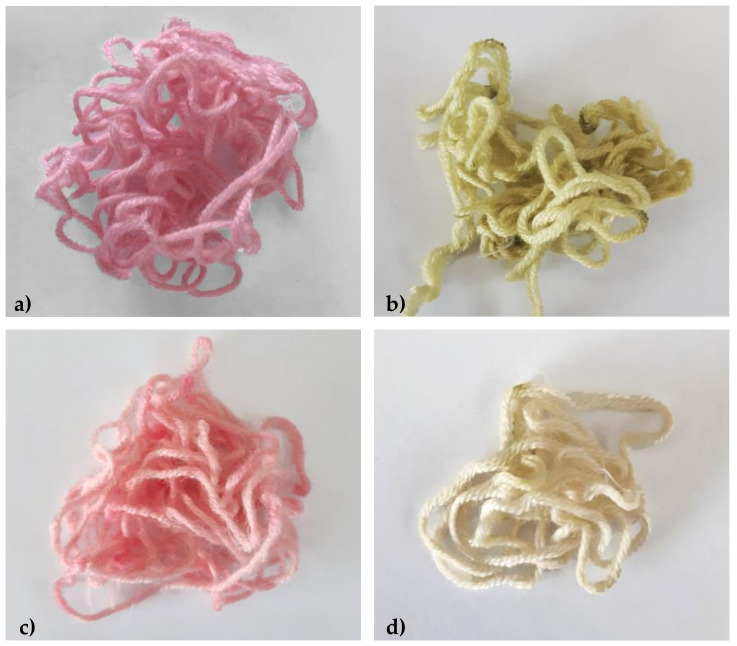
The visual appearance of dyed wool yarn samples before and after washing with a mildly alkaline reference detergent. (**a**) WoolP_1 before washing. (**b**) WoolG_1 after washing. (**c**) WoolP_2 before washing. (**d**) WoolG_2 after washing.

**Table 1 molecules-26-06775-t001:** The pH of mixtures of chicory powder and different extraction solvents.

% Ethanol	% Acid	pH			
		Acetic Acid	Ascorbic Acid	Citric Acid	Tartaric Acid
0%	0.20%	2.85	2.67	2.27	2.12
	2%	2.27	2.1	1.68	1.6
12.50%	0.20%	2.84	2.7	2.27	2.22
	2%	2.4	2.1	1.75	1.68
25%	0.20%	2.96	2.82	2.45	2.25
	2%	2.38	2.3	1.93	1.77
50%	0.20%	3.35	3.23	2.75	2.67
	2%	2.65	2.65	2.14	2.01

**Table 2 molecules-26-06775-t002:** Putative anthocyanins in red chicory identified by UPLC-qTOF-MS.

Peak No. ^1^	RT (Min) *	Ion Mode *	Exp. Mass (m/z) *	Theor. Mass (m/z) *	Error (ppm)	Formula	MS/MS	Putative Identification
1	15.68	Pos	449.1083	449.1084	−0.2	C_21_H_21_O_11_	287.0562	Cyanidin 3-*O*-glucoside
2	18.09	Pos	783.162	783.162	0	C_33_H_35_O_22_	535.1083; 287.0560	Cyanidin 3,5-di-*O*-(6″-*O*-malonyl)-glucoside
2	18.09	Pos	551.103	551.1037	−1.3	C_24_H_23_O_15_	303.0509	Delphinidin 3-*O*-(6″-*O*-malonyl)-glucoside
2	18.09	Neg	505.0985	505.0982	0.6	C_23_H_23_O_13_	300.0271	Delphinidin 3-*O*-(6″-*O*-acetyl)-glucoside
3	19.5	Pos	535.1091	535.1088	0.6	C_24_H_23_O_14_	287.0561	Cyanidin 3-*O*-(6″-*O*-malonyl)-glucoside
3	19.5	Neg	489.1042	489.1033	1.8	C_23_H_23_O_12_	284.033	Cyanidin 3-*O*-(6″-*O*-acetyl)-glucoside
4	21.1	Pos	549.1238	549.1244	−1.1	C_25_H_25_O_14_	287.0564	Cyanidin succinyl hexoside
5	20.1	Pos	643.1303	643.1299	0.6	C_30_H_27_O_16_	287.0553	Cyanidin derivative 1
6	22.44	Pos	563.1401	563.1401	0	C_26_H_27_O_14_	287.0559	Cyanidin derivative 2

^1^ Peak numbers match those shown in Appendix A. * RT = retention time, ion mode = ionization mode, exp. mass = experimentally determined mass; theor. mass = theoretical mass.

**Table 3 molecules-26-06775-t003:** Kinetic parameters for the two dyeing processes WoolP_1 and WoolP_2.

Samples	Pseudo-First-Order Model		Pseudo-Second-Order Model		
	R^2^	k_1_ (/min)	R^2^	q_e_ (mg/g)	k_2_ (g/mg min)
WoolP_1	0.934	−0.0032	0.991	0.48	0.0019
WoolP_2	0.959	−0.0035	0.994	0.44	0.0022

**Table 4 molecules-26-06775-t004:** Fastness analysis of dyed wool yarn samples before and after washing with a mildly alkaline reference detergent.

WOOL_P1	Washing Fastness	Acid Perspiration Fastness	Alkaline Perspiration Fastness	Light Fastness
Sample fading	1	2/3	1	1/2
Staining on wool	5	5	5	
Staining on acrylic	5	5	5	
Staining on polyester	5	5	5	
Staining on polyamide	5	4	4/5	
Staining on cotton	4/5	4	4	
Staining on acetate	5	5	5	
**WOOL_G1**	**Washing fastness**	**Acid perspiration fastness**	**Alkaline perspiration fastness**	**Lightfastness**
Sample fading	4	1	3/4	2
Staining on wool	5	5	5	
Staining on acrylic	5	5	5	
Staining on polyester	5	5	5	
Staining on polyamide	5	5	5	
Staining on cotton	5	4	5	
Staining on acetate	5	5	5	
**WOOL_P2**	**Washing fastness**	**Acid perspiration fastness**	**Alkaline perspiration fastness**	**Lightfastness**
Sample fading	1	1	1	2
Staining on wool	5	5	5	
Staining on acrylic	5	5	5	
Staining on polyester	5	5	5	
Staining on polyamide	5	5	4/5	
Staining on cotton	4/5	4/5	4/5	
Staining on acetate	5	5	5	
**WOOL_G2**	**Washing fastness**	**Acid perspiration fastness**	**Alkaline perspiration fastness**	**Lightfastness**
Sample fading	4	2	4	2
Staining on wool	5	5	5	
Staining on acrylic	5	5	5	
Staining on polyester	5	5	5	
Staining on polyamide	5	5	5	
Staining on cotton	4/5	4/5	5	
Staining on acetate	5	5	5	

## Data Availability

Not applicable.

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
