# Peer review of "Optimization of a Sustainable Protocol for the Extraction of Anthocyanins as Textile Dyes from Plant Materials"

_molecules, 2021, doi:10.3390/molecules26226775_

Round 1

Reviewer 1 Report

The manuscript Optimization of a sustainable protocol for the extraction of anthocyanins as textile dyes from plant materials by Elisa Gecchele et al. describes the study aiming to develop a sustainable and environmentally friendly method of obtaining natural compounds for textile dyeing. This is an important subject with a great potential for application in the industry.

The study is very well designed and the manuscript very well written, it has a logical structure, the results are clearly presented and promising. In my opinion the article can be accepted after some minor changes. However, although it fits the scope of the Molecules, it is a rather specialized topic which would receive better attention in a journal with a narrower scope which is more focused on the research on dyes.

Some minor issues that should be addressed are:

  • The conclusion does not conclude the work, it should be rewritten to reflect on the findings of this study.
  • The chemical names of the compounds are given but the article could use a Figure with at least some major anthocyanines, preferably in the introduction section.
  • You mention that the extracts were decomposing and assessed the decomposition process but did you consider what products may be formed? Were the degraded products colorless and what about their potential biological effects, i.e. toxicity to humans or to aquatic organisms after washing off?
  • The -O- indicating the position of substitution at the oxygen atom in chemical names should be italicized throughout the manuscript (lines 56-59, table 2, lines 224-227, 231-235, 244-248, 370-378, 389).
  • Lines 264, 488, 489 and 497: absorption should be changed to adsorption, while in lines 266 and 491 – absorbed to adsorbed.
  • Lines 509, 519, and 521 – the liter is abbreviated as L, while throughout the manuscript it is abbreviated as l (as in ml).

Author Response

1)The conclusion does not conclude the work, it should be rewritten to reflect on the findings of this study.

We thank the reviewer for the comment and we agree with the suggestion. We modified the conclusion section with a better focus on the findings reported in the article.

2)The chemical names of the compounds are given but the article could use a Figure with at least some major anthocyanines, preferably in the introduction section.

Figure 1 was added to the manuscript illustrating the main chicory anthocyanins reported in the text.

3)You mention that the extracts were decomposing and assessed the decomposition process but did you consider what products may be formed? Were the degraded products colorless and what about their potential biological effects, i.e. toxicity to humans or to aquatic organisms after washing off?

We thank the reviewer for the question. The metabolomic analyses were focused on anthocyanins and the content was assessed through HPLC-DAD and the identities through UPLC-MS. We modified the manuscript by better highlighting this point (rows 192-194).

4)The -O- indicating the position of substitution at the oxygen atom in chemical names should be italicized throughout the manuscript (lines 56-59, table 2, lines 224-227, 231-235, 244-248, 370-378, 389).

Done

5)Lines 264, 488, 489 and 497: absorption should be changed to adsorption, while in lines 266 and 491 – absorbed to adsorbed.

Done.

6)Lines 509, 519, and 521 – the liter is abbreviated as L, while throughout the manuscript it is abbreviated as l (as in ml).

Done.

Reviewer 2 Report

Peer-review considerations regarding the manuscript entitled "Optimization of a sustainable protocol for the extraction of anthocyanins as textile dyes from plant materials" signed by Elisa Gecchele et al. are as follow:

1. Abstract: 

2. Introduction:

  • Comparison between natural dyes, in general, and synthetic dyes as argument for the topic of the manuscript is well done.
  • Anthocyanins as natural dyes extracted from radicchio as well the aim of the study—development of an experimental design for anthocyanins extraction from radicchio—are well presented
  • The references are relevant for the topic of the study.

3. Results:

The results are well presented, correct statistical tests are mentions. The sections and subsection are very well organised and logical. The anthocyanines stability in time as well as molecular characterization is very well done and scientifically corect and complete. However, there are some aspects that are not completely clear or could be improved:

  • While for the incubation time, the authors explain that despite no significance between the time used, they chose the 30 minutes due to low variability, for the LFW/solvent ratio they chose 1:20, although 1:60 was better. They could present the reason why the selected this ratio, probably due to less solvent consumption.
  • Figure 3 has the font of the text so small that could be barely read.
  • Figure 8 has very low resolution. It should be increased for better clarity.
  • Title from section 2.5 isn't better "Colour fastness" instead of "Textile fastness"?

4. Discussions

The authors discuss their results in detail in the order that was also used in the Results section. The discussions are correctly and carefully done in the light of the literature that they cited.

5. Materials and Methods:

This section is well organized and complete. There are some minor aspects that need attention:

  • The formula used in section 4.3 for anthocyanines quantification correctly specify the Abs, however, the absorbance should be registered at a certain wavelength not a large domain (400-600 nm). What was the wavelength used (was it 520 nm as stated in section 4.8.2?, if so need to be specified)? Moreover the indicated extinction coeficient is at certain wavelength, what is the wavelength?
  • Section 4.6 need to include the chromatographic column and the chromatographic protocol.

Conclusions are supported by the presented results.

Author Response

Peer-review considerations regarding the manuscript entitled "Optimization of a sustainable protocol for the extraction of anthocyanins as textile dyes from plant materials" signed by Elisa Gecchele et al. are as follow:

  1. Abstract: 
  2. Introduction:
  • Comparison between natural dyes, in general, and synthetic dyes as argument for the topic of the manuscript is well done.
  • Anthocyanins as natural dyes extracted from radicchio as well the aim of the study—development of an experimental design for anthocyanins extraction from radicchio—are well presented
  • The references are relevant for the topic of the study.
  1. Results:

The results are well presented, correct statistical tests are mentions. The sections and subsection are very well organised and logical. The anthocyanines stability in time as well as molecular characterization is very well done and scientifically corect and complete. However, there are some aspects that are not completely clear or could be improved:

  • While for the incubation time, the authors explain that despite no significance between the time used, they chose the 30 minutes due to low variability, for the LFW/solvent ratio they chose 1:20, although 1:60 was better. They could present the reason why the selected this ratio, probably due to less solvent consumption.

We thank the reviewer for the comment, we used 1:20 for less solvent consumption and we highlight this in the text (rows:89-90).

  • Figure 3 has the font of the text so small that could be barely read.

We thank the reviewer for the suggestion and we modified the font size accordingly

  • Figure 8 has very low resolution. It should be increased for better clarity.

The resolution of Figure 8 has been increased as suggested.

  • Title from section 2.5 isn't better "Colour fastness" instead of "Textile fastness"?

The title has been corrected in the text

  1. Discussions

The authors discuss their results in detail in the order that was also used in the Results section. The discussions are correctly and carefully done in the light of the literature that they cited.

  1. Materials and Methods:

This section is well organized and complete. There are some minor aspects that need attention:

The formula used in section 4.3 for anthocyanines quantification correctly specify the Abs, however, the absorbance should be registered at a certain wavelength not a large domain (400-600 nm). What was the wavelength used (was it 520 nm as stated in section 4.8.2?, if so need to be specified)? Moreover the indicated extinction coeficient is at certain wavelength, what is the wavelength?

We thank to reviewer for this comment. We used a 520 nm wavelength and we modified the text by clarifying this aspect.

Section 4.6 need to include the chromatographic column and the chromatographic protocol.

Done. We have modified the section 4.6 accordingly.

Conclusions are supported by the presented results.

Reviewer 3 Report

Throughout the work, I did not understand what the aim of the work was. 1. Line 64-68 need to be reformulated into a work aim.
2.
The work should be written in the infinitive so the whole work should be corrected.
3. It was written in Chapter 4.9 - All assays were carried out at least three times and the data are presented as means 535
± standard deviations.....
where then is the statistical analysis of Figure 1 - Figure 9 and Table 1 - Table 4.
4. In Chapter 3.
there are almost no comparisons of the results with other authors and their research.
5. Why arent the norms according to wich the analyzes were performed listed in Chapter 4 Materials and Methods?
Why aren't the norms according to which the analyzes were performed listed in Chapter 4? Why no norm according to which the analyzes were performed is cited in Chapter 4 Why no norm according to which the analyzes were performed is cited in Chapter 4 Why no norm according to which the analyzes were performed is cited in Chapter 4

Author Response

Throughout the work, I did not understand what the aim of the work was. Line 64-68 need to be reformulated into a work aim.

We modified the text accordingly to reviewer’s suggestion (Lines64-68).

The work should be written in the infinitive so the whole work should be corrected.

We do not understand the reviewer’s recommendation that the article “should be written in the infinitive”. Restricting the entire article to the infinitive would rule out the use of finite verbs, participles and gerunds, and would make the language extremely unnatural and awkward. We also feel that this comment moves beyond objective criticism of the research and its presentation and more into the domain of the reviewer’s subjective personal preference of writing styles. This is outside the scope of the reviewer’s role and we respectfully disagree with the opinion.

 It was written in Chapter 4.9 - All assays were carried out at least three times and the data are presented as means ± standard deviations..... where then is the statistical analysis of Figure 1 - Figure 9 and Table 1 - Table 4.

We confirm that all the experiments were carried out using at least three replicates. The results were then reported as means ± standard deviations.

Data shown in Figure 1 are reported as means ± SD obtained from three independent experiments and then subjected to one-way ANOVA statistical analysis with Tukey’s post hoc test (p<0.05), as explained in the figure caption. Significant differences resulting from this analysis are indicated in the graphs by different letters, as indicated.

Figure 9 is a schematic representation of the protocols for the dyeing of woolen yarn with plant-derived anthocyanins on weight fibers in the presence or absence of potassium alum as a mordant. Then, no statistical analysis can be performed on  such an experimental procedure.

In table 1 the pH of the different extraction solvents tested in the DoE approach to select the best extraction buffer are reported. In this case the aim was not to statistically compare the pH values, but to highlight the differences in pH values depending on the acid organic used.

In table 4 results obtained from the fastness analysis of dyed wool yarn samples before and after washing with a mildly alkaline reference detergent are reported. No statistical approach can be performed for this data setting.

 In Chapter 3. there are almost no comparisons of the results with other authors and their research.

We compared our results with the ones present in literature by Lavelli et al and Migliorini et al (rows:371-373).

Why arent the norms according to which the analyzes were performed listed in Chapter 4 Materials and Methods? Why aren't the norms according to which the analyzes were performed listed in Chapter 4? Why no norm according to which the analyzes were performed is cited in Chapter 4 Why no norm according to which the analyzes were performed is cited in Chapter 4Why no norm according to which the analyzes were performed is cited in Chapter 4

We do not understand the request/s (?) of the reviewer.